# HLogformer: A Hierarchical Transformer for Representing Log Data

## Abstract

Transformers have gained widespread acclaim for their versatility in handling diverse data structures, yet their application to log data remains underexplored. Log data, characterized by its hierarchical, dictionary-like structure, poses unique challenges when processed using conventional transformer models. Traditional methods often rely on manually crafted templates for parsing logs, a process that is labor-intensive and lacks generalizability. Additionally, the linear treatment of log sequences by standard transformers neglects the rich, nested relationships within log entries, leading to suboptimal representations and excessive memory usage.

To address these issues, we introduce *HLogformer*, a novel hierarchical transformer framework specifically designed for log data. *HLogformer* leverages the hierarchical structure of log entries to significantly reduce memory costs and enhance representation learning. Unlike traditional models that treat log data as flat sequences, our framework processes log entries in a manner that respects their inherent hierarchical organization. This approach ensures comprehensive encoding of both fine-grained details and broader contextual relationships. Our contributions are threefold: First, *HLogformer* is the first framework to design a dynamic hierarchical transformer tailored for dictionary-like log data. Second, it dramatically reduces memory costs associated with processing extensive log sequences. Third, comprehensive experiments demonstrate that *HLogformer* more effectively encodes hierarchical contextual information, proving to be highly effective for downstream tasks such as synthetic anomaly detection and product recommendation.

## 1 Introduction

In recent years, transformers have garnered significant attention due to their versatility in handling various data structures, including images, text, graphs, tabular data, and temporal graphs (Vaswani et al., 2017; Dosovitskiy et al., 2020; Veličković et al., 2017; Huang et al., 2020; Wu et al., 2024; Hou et al., 2024b). Despite their widespread application, there remains a notable gap in research focused on log data. Log data inherently possesses a hierarchical, dictionary-like structure, where each log entry is composed of nested fields and attributes. For instance, a single log entry might include metadata like timestamps, user IDs, and event types at the top level, while containing nested details such as specific actions taken, resources affected, and additional contextual information. Examples of log data include Amazon EC2 logs, IAM logs, and web server access logs.

Traditional methods for processing log data often involve manually applying templates to parse the logs before utilizing existing transformers. These templates are predefined rules or patterns designed to extract structured information from unstructured log messages. While this approach can be effective for certain types of logs, it has several limitations. Template-based methods can be labor-intensive, requiring domain-specific knowledge to create and maintain the templates. Additionally, they may not generalize well to diverse or evolving log formats, leading to incomplete or inaccurate parsing.

When lengthy log sequences are input into transformers for representation learning and downstream tasks, several challenges arise. Firstly, the memory requirements become excessive due to the sheer volume of log data, making it difficult to process efficiently. Secondly, capturing the necessary contextual information demands larger and more complex transformer models, which can be com-

putationally expensive and resource-intensive. Lastly, there is a tendency to treat log data as linear sequences, which neglects the hierarchical and structured nature of log entries. This linear treatment fails to leverage the rich, nested relationships inherent in log data, resulting in sub-optimal representation and analysis.

To address these challenges, researchers have proposed several approaches aimed at extending context length and reducing memory costs. Sparse transformers (Child et al., 2019) leverage predefined patterns to limit the number of attention connections each token has. Local attention restricts the attention mechanism to a fixed-size window around each token, ensuring that only nearby tokens are considered. This approach is efficient for capturing local dependencies and reduces the overall computational burden. Strided attention extends this idea by allowing tokens to attend to other tokens at fixed intervals, further reducing the number of attention connections while maintaining the ability to capture broader context across the sequence. Other methods, such as the ones proposed by Roy et al. (2021) and Kitaev et al. (2020), take this concept further by making the sparsity pattern learnable.

Additionally, models like Longformer (Beltagy et al., 2020), ETC (Extended Transformer Construction) (Ainslie et al., 2020), and Big Bird (Zaheer et al., 2020) introduce global memory tokens to address the limitations of traditional transformers in handling long sequences. These global memory tokens are specialized tokens that have attention connections to all other tokens in the sequence. This mechanism enables the models to maintain a broader contextual understanding without the quadratic memory and computational overhead typically associated with the self-attention mechanism in standard transformers. There are techniques such as Transformer-XL (Dai et al., 2019) and Compressive Transformer (Rae et al., 2019) which employ segment-based recurrence to significantly reduce memory and computational costs. Despite their effectiveness, these approaches are not tailored to the unique characteristics of log data.

There are several hierarchical transformers (Nawrot et al., 2021; Pappagari et al., 2019; Pan et al., 2021; Liu et al., 2021b) that modify the vanilla transformer architecture to obtain hierarchical representations of the data. However, these architectures primarily build the hierarchy by encoding the tokens using downsampling, pooling, or segmentation techniques, which are not specifically designed for the hierarchical log data we are interested in.

In this paper, we introduce a novel and efficient hierarchical transformer framework specifically designed for log data, termed *HLogformer*. Our *HLogformer* framework addresses the unique challenges of log data by significantly reducing memory costs, making it feasible to apply transformers to lengthy log sequences. Furthermore, *HLogformer* captures and leverages the inherent hierarchical structural information within the data, thereby enhancing representation learning. Our key contributions are as follows:

- *HLogformer* is the first framework to design a dynamic hierarchical transformer tailored for dictionary-like nested log data.

- *HLogformer* dramatically reduces memory costs associated with processing extensive log data.

- Comprehensive experiments demonstrate that *HLogformer* more effectively encodes hierarchical contextual information, proving to be highly effective for downstream tasks such as synthetic anomaly detection and product recommendation.

The rest of the paper is organized as follows. Section 2 reviews the related work, providing context and background that underpins our study. In Section 3, we delve into the proposed methodology and training strategy, detailing the innovative approaches and techniques we employ. Finally, Section 4 presents the experiments and results, showcasing the effectiveness and practical implications of our proposed model.

## 2 RELATED WORKS

The related work in this area can be categorized into 2 main groups: efficient transformers including incorporating global memory tokens, sparse attention mechanisms, segment-based recurrence methods, and hierarchical architectures. Each category offers distinct approaches to addressing the challenges of processing long sequences with transformers.

## 2.1 EFFICIENT TRANSFORMERS

**Global Memory Tokens in Transformers.** Models like Longformer (Beltagy et al., 2020), ETC (Extended Transformer Construction) (Ainslie et al., 2020), and Big Bird (Zaheer et al., 2020) introduce global memory tokens to address the limitations of traditional transformers with long sequences. These tokens maintain attention connections to all other tokens in the sequence, allowing the models to capture broader contextual understanding while avoiding the quadratic memory and computational overhead of standard self-attention mechanisms.

**Sparse Attention Mechanisms.** Sparse transformers (Child et al., 2019) employ fixed patterns with local and strided attention to address the inefficiencies of traditional transformers in processing long sequences. Other methods, such as those proposed by (Roy et al., 2021) and (Kitaev et al., 2020), enhance this concept by making the sparsity pattern learnable. These approaches adapt the attention patterns during training to better capture the data structure.

**Segment-based Recurrence.** Segment-based recurrence methods, such as Transformer-XL (Dai et al., 2019) and Compressive Transformer (Rae et al., 2019), introduce mechanisms to maintain and leverage contextual information across segments, significantly reducing memory and computational costs.

Despite their effectiveness, these approaches are not specifically tailored to the unique characteristics of log data, which often exhibit a hierarchical, dictionary-like structure. This gap underscores the need for models designed to capture and leverage the intrinsic structure of log data.

## 2.2 HIERARCHICAL ARCHITECTURES

Existing hierarchical transformer architectures (Nawrot et al., 2021; Pappagari et al., 2019; Pan et al., 2021; Liu et al., 2021b; He et al., 2021) that primarily focus on compressing or encoding fine-grained information and decoding it back to the original size if necessary. For example, Hourglass (Nawrot et al., 2021) utilizes downsampling and upsampling techniques to create hierarchical and efficient transformers. Pappagari et al. (2019) design hierarchical transformers by segmenting the input into smaller chunks and feeding each chunk into the base model, effectively managing long documents. Swin Transformer (Liu et al., 2021b) employs a shifted windows scheme to design an efficient hierarchical architecture. sentence-level information in text data. However, these architectures often prioritize compression and encoding efficiency over accurately representing the hierarchical nature of data. They focus on reducing the size of the data for efficient processing and storage, and then decoding it back when needed. These approaches do not fully align with the unique characteristics of log data, which require capturing and leveraging their inherent hierarchical structure.

## 2.3 TRUSTWORTHINESS IN LANGUAGE MODELING

Trustworthiness in language modeling attract more attention in recent years (Morris et al., 2020; Tao et al., 2024a;b; Luo et al., 2024). For instance, Li et al.Li et al. (2020) enhance text-to-speech transformers by modifying attention and position embedding. Liu et al.(Liu et al., 2021a) introduce an attention-based classifier for crisis detection. TableFormer (Yang et al., 2022) improves tabular data encoding for robustness. However, these models are task-specific and lack generalizability. Han et al. (Han et al., 2023) address this by proposing a general self-attention framework using robust kernel density estimation (RKDE).

## 3 PROPOSED METHODOLOGY

In this section, we discuss the hierarchical structure inherent in log data and introduce our novel model, *HLogformer*, designed to leverage this structure.

### 3.1 HIERARCHICAL STRUCTURE OF LOG DATA

As illustrated in Figure 1 log data, such as AWS CloudTrail Logs, can be represented in two distinct ways: as a linear sequence (Figure 1 (a)) or as a hierarchical tree (Figure 1 (b)).

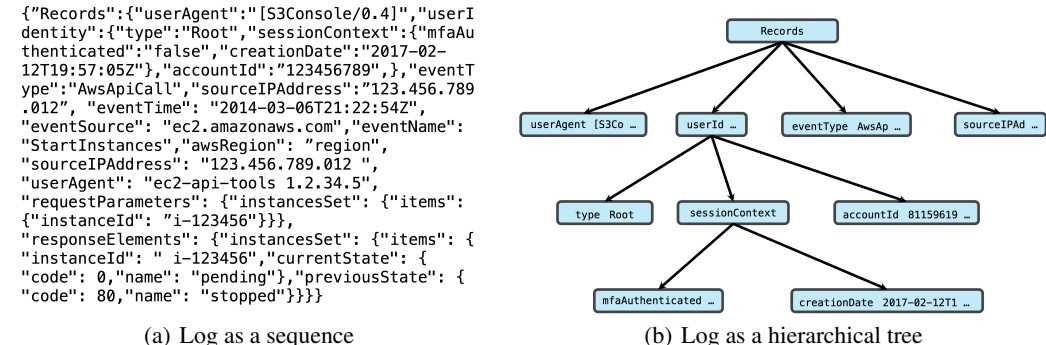

{"Records":{"userAgent":"[S3Console/0.4]","userI dentity":{"type":"Root","sessionContext":{"mfaAu thenticated":"false","creationDate":"2017-02-12T19:57:05Z"},"accountId":"123456789",},"eventT ype":"AwsApiCall","sourceIPAddress":"123.456.789 .012", "eventTime": "2014-03-06T21:22:54Z", "eventSource": "ec2.amazonaws.com","eventName": "StartInstances","awsRegion": "region", "sourceIPAddress": "123.456.789.012 ", "userAgent": "ec2-api-tools 1.2.34.5", "requestParameters": {"instancesSet": {"items": {"instanceId": "i-123456"}}}, "responseElements": {"instancesSet": {"items": { "instanceId": " i-123456","currentState": { "code": 0,"name": "pending"},"previousState": { "code": 80,"name": "stopped"}}}}

(a) Log as a sequence                    (b) Log as a hierarchical tree

Figure 1: Different representations of log data: (a) treating log data as a sequence, and (b) treating the log data as a hierarchical tree.

When log data is represented as a sequence (Figure 1 (a)), each log entry is treated as a part of a continuous stream. This sequential representation allows for the application of traditional language modeling techniques, where each log entry is analogous to a token in a sentence. By leveraging vanilla language models it is possible to derive meaningful representations of the log data.

However, treating log data as a sequence can oversimplify the complex, nested relationships inherent in the logs. Each log entry in systems like CloudTrail contains multiple fields and attributes organized in a hierarchical structure, reflecting the nested nature of the recorded events. For example, user identity as a log entry contains nested attributes such as account Id, username, session context, principal Id, where session context itself has a nested structure and contains attributes such as session issuer, session arn, etc. Representing this data as a flat sequence can obscure these relationships and result in a loss of critical contextual information.

Representing log data as a hierarchical tree (Figure 1 (b)) acknowledges and preserves the nested structure of the log entries. In this representation, each node in the tree corresponds to a component of the log entry, with parent-child relationships reflecting the inherent hierarchy. This approach captures the multi-level dependencies and relationships within the data more effectively, allowing for a richer and more accurate representation.

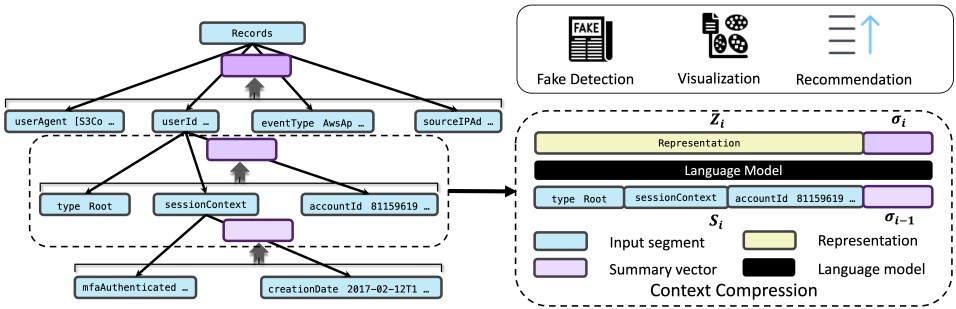

Figure 2: Schematic overview of HLogformer: HLogformer encapsulates the context segment into a summary vector, which is then passed from low-level to high-level (left). Specifically, at each step, we concatenate all the child nodes' tokens $S_i$ and the previous summary vector $\sigma_{i-1}$ as the input. The language model is then applied over this input to obtain the updated summary vector and the token representation (right).

## 3.2 HLOGFORMER: A HIERARCHICAL LOG TRANSFORMER

To fully leverage the hierarchical structure inherent in log data, we introduce a novel architecture called *HLogformer*, illustrated in Figure 2 . This architecture is inspired by context compression techniques (Chevalier et al., 2023), but unlike them, *HLogformer* segments log data according to its hierarchical tree structure. This segmentation process progresses systematically from low-level

details to high-level summaries, mirroring the natural organization of the data. Each segment corresponds to a distinct level of the hierarchical structure, ensuring that the model respects and utilizes the nested relationships within the log entries.

We can first represent the log data as a directed graph $\mathcal{G} = (\mathcal{V}, \mathcal{E})$ where $s_i$ denotes the text in node $v_i \in \mathcal{V}$ while $e_{ij} = (v_i, v_j) \in \mathcal{E}$ denotes the parent-child relationship in the log data. For step $i$, we concatenate all the child nodes' text of node $i$ as the segment $S_i = \textbf{Concat}[\{s_j : e_{ij} \in \mathcal{G}\}]$.

The processing pipeline of *HLogformer* operates step-by-step as shown in Figure 2 (right), beginning with the most granular details of the log data. At each step, the architecture processes a segment of the log data, extracting and summarizing the relevant information. These summary vectors encapsulate the essential context and dependencies at the current level of the hierarchy. Once processed, these summary vectors are passed to the next step, where higher-level segments are processed similarly. At each step $i$, the segment $S_i$ is processed along with the summary vector from the previous step $\sigma_{i-1}$. This process ensures that the hierarchical context is preserved and progressively refined as we move through the log data. The following equation formalizes this process, where the log data segment $S_i$ and the summary vector from the previous step $\sigma_{i-1}$ are combined and processed by the language model **LM**:

$$Z_i, \sigma_i = \textbf{LM}([S_i, \sigma_{i-1}]) \tag{1}$$

In this equation, **LM** represents the language model that generates the new summary vector $\sigma_i$ and the intermediate representation $Z_i$, capturing both the current segment's information and the accumulated context from previous segments.

**Bidirectional Hierarchical Compression Paradigm.** In the primary architecture described above, summary vectors are passed exclusively from low-level to high-level segments. This allows high-level tokens to access low-level information through the summary vectors, but it may result in low-level tokens missing out some corresponding high-level context. To address this limitation, we propose a bidirectional summary passing technique. This involves initially passing the summary from low-level to high-level, and then reversing the process to ensure that low-level tokens can also benefit from high-level information.

**Complexity Analysis.** Our *HLogformer* provides an efficient framework for handling long context in log data. Assume the entire sequence has a length of $L$ and is split into $M$ equal-sized segments. Then the vanilla transformer has a memory complexity of $O(L^2)$, while *HLogformer* reduces this to $O(L^2/M)$.

**Advantages.** This progressive approach offers several key advantages: (1) By segmenting the log data according to its hierarchical structure, *HLogformer* captures both fine-grained details and broader contextual relationships, building a comprehensive and layered representation at each step; (2) This method significantly reduces memory and computational costs by summarizing information at each level and passing only the accumulated summary vectors to the next step, efficiently managing the data's complexity and size; (3) Additionally, *HLogformer* enhances the model's ability to perform downstream tasks such as anomaly detection, log classification, and predictive maintenance. By maintaining and leveraging the hierarchical structure, the model can more accurately identify patterns and anomalies within the data.

### 3.3 TRAINING STRATEGY

After building the hierarchical log transformer, we need to adopt an appropriate training strategy to obtain informative representations and perform downstream tasks. Given that log data typically lack labels, we propose a *self-supervised learning approach* using masked language modeling loss and volume hypersphere minimization loss, as illustrated in Figure 3.

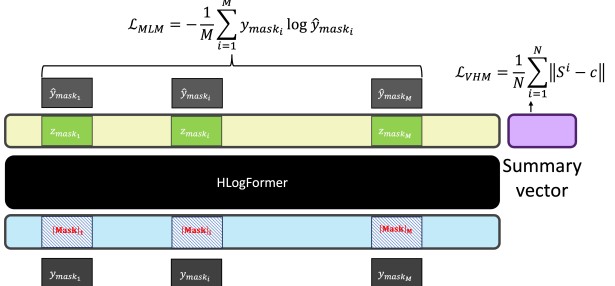

Figure 3: Self-supervised Learning.

**Masked Language Modeling.** To capture the contextual information of log data, we utilize the masked language modeling (MLM) task, which has proven effective in various natural language processing applications. This approach involves randomly selecting a subset of tokens from the input data and replacing them with a special `[MASK]` token. The model is then tasked with predicting the original tokens that were masked, allowing it to learn rich contextual representations of the log data.

The training objective for this task is defined by the cross-entropy loss function, which measures the discrepancy between the predicted tokens and the actual tokens at the masked positions. Formally, the MLM loss is expressed as:

$$\mathcal{L}_{MLM} = \frac{1}{M} \sum_{i=1}^{M} y_{mask_i} \log \hat{y}_{mask_i},$$

where $M$ is the number of masked tokens, $y_{mask_i}$ represents the actual token at the $i$-th masked position, and $\hat{y}_{mask_i}$ is the predicted token at the same position. This loss function encourages the model to accurately predict the masked tokens, thereby forcing it to learn the underlying patterns and dependencies in the log data.

**Volume Hypersphere Minimization.** Given our assumption that all training data represents real or normal instances, the task aligns well with one-class classification problems. In this context, we draw inspiration from the One-Class Deep SVDD (Ruff et al., 2018) methodology. Our objective is to map normal data points as closely as possible to the center of a hypersphere. This approach effectively captures the notion of normality by ensuring that the representations of normal data points are densely clustered.

To achieve this, we seek to minimize the volume of the hypersphere by positioning its center, denoted as $c$, such that the mean distance of all data representations to this center is minimized. Formally, this minimization problem is expressed through the following loss function:

$$\mathcal{L}_{VHM} = \frac{1}{N} \sum_{i=1}^{N} \|S^i - c\|,$$

where $N$ is the number of data points, $S^i$ represents the accumulated summary vector of $i$-th data point, and $c = \frac{1}{N} \sum_{i=1}^{N} S^i$ is the calculated center of all the data representations. This center $c$ is dynamically computed as the average of all summary vectors, ensuring that it accurately reflects the central tendency of the normal data points.

By minimizing this loss, we encourage the model to produce representations that are not only compact but also concentrated around a central point. This is crucial for downstream tasks such as anomaly detection, where deviations from this central cluster can be effectively identified as anomalies.

## 4 EXPERIMENTS

In this section, we will begin by evaluating the effectiveness of our *HLogformer* model in masked language modeling tasks. This will involve assessing its ability to accurately predict masked tokens within a sequence, thereby demonstrating its understanding of the underlying log data. Following this, we will apply our model to several downstream tasks to further validate its utility and performance. These tasks include fake log detection, where the model will be tested on its capability to identify fraudulent or synthetic log entries, and visualization analysis, where we will leverage the model's outputs to generate insightful visual representations of the log data.

### 4.1 EXPERIMENTAL SETTING

In this section, we detail the dataset utilized for our experiments, the backbone architecture underpinning our models, and the hyperparameters selected to optimize performance.

**Datasets.** We use the following datasets in our experiments:

(1) CloudTrail Logs Dataset: It is an anonymized public log data from `flaws.could` that covers over 3.5 years of data and 1,939,207 number of events.

(2) OKTA: This log data is a private dataset which monitors and audits authentication activity to an internal system.

(3) TrailDiscover: It is an evolving repository of CloudTrail events with detailed descriptions, MITRE ATT&CK insights, real-world incidents, references and security implications.

(4) Amazon Reviews (Hou et al., 2024a): It is collected in 2023 by McAuley Lab. We use 9 categories of Item Metadata in Amazon Reviews including All Beauty, Amazon Fashion, Appliances, Arts Crafts and Sewing, Automotive, CDs and Vinyl, Digital Music, Health and Personal Care, and Magazine Subscriptions.

**Backbone Architectures.** Since our *HLogformer* is designed as a versatile plugin capable of integrating with any transformer architecture, we will experiment with a variety of backbone models to demonstrate its adaptability and effectiveness. Specifically, we will employ several transformer architectures, including the vanilla Transformer (Devlin et al., 2018) with random initialization, pretrained Transformer (Devlin et al., 2018) with pretrained parameters in `bert-base-uncase` and four efficient transformers: Linear Transformer (Katharopoulos et al., 2020), Reformer (Kitaev et al., 2020), Routing Transformer (Roy et al., 2021), and Sparse Transformer (Child et al., 2019).

**Hyperparameters.** We use 8 transformer blocks for backbone models and 1 block for our *HLogformer*. We set the number of training epochs to 100, the masking rate to 0.2, and the length of the summary vector to 10 tokens. We use Adam optimizer with the learning rate of {0.01,0.005, 0.001 }, the Adam weight decay of {0.01, 0.001,0.0001}, Adam $\beta_1$ of {0.3,0.6, 0.9}, and Adam $\beta_2$ of { 0.9, 0.99, 0.999}. For each dataset, the training/validation/testing ratio is set as 5:1:1.

## 4.2 Self-Supervised Learning Task

To demonstrate the effectiveness and efficiency of our HLogformer, we train the models with masked language modeling loss and present the loss and the number of parameters in the transformer block in Table 1 (Security datasets) and Table 2 (Amazon Reviews datasets). From these tables, we can make the following observations:

- Our hierarchical framework is a highly effective plug-in module that significantly and consistently reduces masked language modeling loss and therefore improves the ability to capture contextual information.

- Our *HLogformer* requires only a small-sized transformer block while achieving better results than the backbone models. As we mentioned in the hyperparameters section, we use only 1 transformer block in *HLogFormer* to handle the segments at each step, while the backbone models require 8 blocks to be able to process the large log data.

| Architecture | CloudTrail | OKTA | TrailDiscover | #Parameter |
|---|---|---|---|---|
| Vanilla Transformer | 5.692 | 4.221 | 5.676 | 12636160 |
| Vanilla-*HLogformer* (Ours) | 4.158 | 2.888 | 4.921 | 789760 |
| Pretrained Transformer | 4.414 | 3.872 | 5.078 | 12636160 |
| Pretrained-*HLogformer* (Ours) | 3.850 | 2.611 | 4.995 | 789760 |
| Linear Transformer | 5.341 | 4.449 | 5.786 | 12636160 |
| Linear-*HLogformer* (Ours) | 4.092 | 2.833 | 5.101 | 789760 |
| Reformer | 4.184 | 3.504 | 5.460 | 13602048 |
| Reformer-*HLogformer* (Ours) | 4.106 | 3.215 | 5.004 | 8537856 |
| Routing Transformer | 6.937 | 5.313 | 9.716 | 10522624 |
| Routing-*HLogformer* (Ours) | 4.186 | 2.748 | 5.323 | 1315328 |
| Sparse Transformer | 8.421 | 5.446 | 8.871 | 4212736 |
| Sparse-*HLogformer* (Ours) | 4.766 | 3.789 | 5.508 | 526592 |

Table 1: Masked language modeling loss on security datasets.

| | Beauty | Fashion | Appliances | Arts | Auto | CDs | Music | Health | Magazine |
|---|---|---|---|---|---|---|---|---|---|
| Vanilla Transformer | 4.571 | 4.82 | 5.029 | 5.267 | 5.018 | 4.345 | 4.316 | 5.021 | 4.054 |
| Vanilla-*HLogformer* (Ours) | 3.686 | 3.46 | 3.999 | 4.372 | 4.57 | 3.449 | 3.565 | 3.936 | 3.159 |
| Linear Transformer | 4.690 | 4.668 | 5.078 | 5.124 | 6.176 | 4.208 | 4.360 | 5.023 | 4.126 |
| Linear-*HLogformer* (Ours) | 3.758 | 3.871 | 3.695 | 4.285 | 4.841 | 3.425 | 3.499 | 3.839 | 2.963 |
| Reformer | 4.069 | 4.212 | 4.389 | 4.581 | 4.741 | 3.545 | 3.978 | 4.349 | 3.283 |
| Reformer-*HLogformer* (Ours) | 3.593 | 4.003 | 4.014 | 4.095 | 4.150 | 3.386 | 3.540 | 3.976 | 2.785 |
| Routing Transformer | 8.494 | 8.503 | 7.871 | 8.771 | 8.665 | 7.454 | 7.532 | 8.561 | 7.400 |
| Routing-*HLogformer* (Ours) | 3.773 | 3.955 | 4.196 | 4.473 | 4.470 | 3.538 | 3.521 | 4.173 | 3.253 |
| Sparse Transformer | 9.680 | 7.654 | 7.470 | 10.196 | 9.666 | 8.629 | 9.561 | 9.567 | 8.687 |
| Sparse-*HLogformer* (Ours) | 4.127 | 3.994 | 5.071 | 4.967 | 4.798 | 3.718 | 4.147 | 4.611 | 3.837 |

Table 2: Masked language modeling loss on Amazon Review datasets.

### 4.3 SUPERVISED LEARNING TASK ON TRAILDISCOVER

In addition to self-supervised learning, we also perform experiments on a supervised classification task. We utilize the TrailDiscover dataset which contains two features for each data point: `"usedInWild"` which is a binary feature and takes two values of True or False, and `"MITRE Attack Tactics"` which is a feature that takes ten different values of attack type. Task 1 is the binary classification task on `"usedInWild"` and Task 2 is the multiclass classification task on `"MITRE Attack Tactics"` . The experimental results in Table 3 show the significant improvement of our *HLogformer* over the backbone transformers.

| Architecture | Task 1 | Task 2 |
|---|---|---|
| Vanilla Transformer | 67.059 | 69.412 |
| Vanilla-*HLogformer* (Ours) | 95.294 | 77.647 |
| Linear Transformer | 65.882 | 51.765 |
| Linear-*HLogformer* (Ours) | 92.941 | 57.647 |
| Reformer | 65.882 | 70.588 |
| Reformer-*HLogformer* (Ours) | 64.706 | 77.647 |
| Routing Transformer | 83.529 | 75.294 |
| Routing-*HLogformer* (Ours) | 90.588 | 78.824 |
| Sparse Transformer | 69.412 | 35.294 |
| Sparse-*HLogformer* (Ours) | 72.941 | 38.823 |

Table 3: Supervised classification task.

### 4.4 SYNTHETIC ANOMALY DETECTION

After conducting the self-supervised training, we obtain the representation of log data as well as the summary vector. Since we assume the model is trained with real data, fake data is likely to exhibit a different distribution or representation pattern compared to real data. Motivated by this, we can utilize the representations and summary vectors to perform fake data detection. To construct the fake dataset, we mismatch the key-value pairs in the real data with a probability of $p = 0.2$. In this section, we divide the fake detection into three parts: (1) detection by loss, (2) detection by fake rate, and (3) detection by visualization.

**Detection by loss.** In this experiment, we train the model with the total loss as $\mathcal{L}_{MLM} + 0.1 \cdot \mathcal{L}_{VHM}$. As we train with the real data, we expect the MLM and VHM losses for real and fake data to show significant differences. Our results in Table 4 demonstrate that the losses for fake data are significantly higher than those for real data. This indicates that self-supervised learning effectively captures the hierarchical context information of data.

| Data | CloudTrail | OKTA | TrailDiscover |
|---|---|---|---|
| Real | 3.925/0.575 | 3.379/0.578 | 4.580/1.833 |
| Fake | 5.841/1.540 | 4.076/1.195 | 5.487/2.738 |

Table 4: Synthetic anomaly detection by MLM/VHM loss.

**Detection by fake rate.** For each masked token $i$, we obtain an output probability $\hat{y}_{mask_i}$. We then construct a candidate set Candidate$_i$ with the top $T$ highest likelihoods. If the real value $x_{mask_i} \in$ Candidate$_i$, we consider token $i$ as normal; otherwise, it is considered fake. Therefore, the fake rate can be calculated as:

$$\text{Fake Rate} = \frac{\text{number of fake tokens}}{\text{number of all masked tokens}} \times 100\%.$$

We leave the detailed results with various $T$ and threshold $\alpha$ in Appendix A.3 due to the space limit.

**Detection by visualization.** With VHM loss, we expect the summary vector of real data to be closely mapped to the center of the hypersphere. Consequently, the representations of real and fake

data should exhibit significantly different patterns. To validate this, we use locally linear embedding (LLE) (Roweis & Saul, 2000), principal component analysis (PCA) and t-distributed stochastic neighbor embedding (t-SNE) (Van der Maaten & Hinton, 2008) to perform the dimension reduction and visualization for the summary vectors obtained from real and fake data. The following figures in Figure 4 demonstrate that the representations of these two data sources are separable and form distinct clusters.

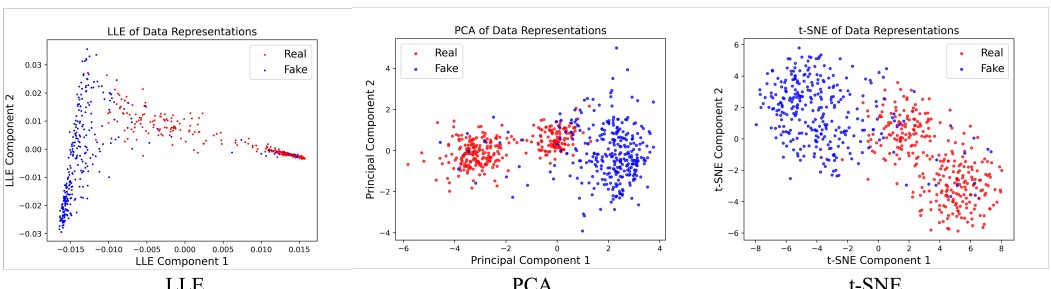

| LLE | PCA | t-SNE |

Figure 4: Visualization of summary vectors.

## 4.5 PRODUCT RECOMMENDATION TASK

To further demonstrate the effectiveness and advantages of our proposed *HLogformer*, we conduct a product recommendation downstream task using the pretrained representations on Amazon Reviews dataset (Hou et al.,

| Precision-$K$ (%) | 1 | 3 | 5 | 8 | 10 |
|---|---|---|---|---|---|
| **Transformer** | 83.50 | 80.67 | 76.10 | 70.81 | 66.65 |
| *HLogformer* | 94.50 | 89.17 | 81.90 | 74.37 | 69.45 |

Table 5: Average precision at different $K$.

2024a). Specifically, we select 200 users with the highest number of purchased items and collect pretrained embeddings for all these items. For each user, the last 10 items purchased are treated as positive samples, while 10 items randomly selected from the available item repository are treated as negative samples. The average embedding of the remaining items is computed to represent the user's embedding. We then calculate the cosine similarity between each item's embedding (both positive and negative) and the user's embedding to generate a score list. This score list is sorted, and precision at $K$ (precision-$K$) is computed based on the top $K$ scores. Finally, we report the average precision-$K$ across all users, as shown in Table 5. The results demonstrate a significant and consistent advantage of our *HLogformer* over the vanilla transformer across all the $K$.

## 4.6 ABLATION STUDIES

To evaluate the effectiveness of our HLogformer, we conduct ablation studies on all of the components. We report the MLM loss in the Table 6, and the results show the effectiveness of all the components in our *HLog-former*.

| Architecture | CloudTrail | OKTA | TrailDiscover |
|---|---|---|---|
| *HLogformer* | 3.850 | 2.611 | 4.995 |
| w/o pretrained | 4.158 | 2.888 | 4.921 |
| w/o hierarchy | 5.692 | 3.269 | 5.676 |
| w/o bi-direction | 4.857 | 4.221 | 5.194 |
| w/o summary | 4.388 | 3.081 | 5.131 |

Table 6: Ablation studies.

## 5 CONCLUSION

In this paper, we propose a novel and efficient hierarchical log transformer for dictionary-like log data. Our hierarchical transformers are specifically designed for log data such as CloudTrail and employ an adaptively recursive architecture tailored to this data. Our hierarchical framework is universal, making it orthogonal and compatible with various transformer backbones to further enhance performance and efficiency. Furthermore, our preliminary experiments show that the hierarchical representation learned through self-supervised learning exhibits great potential for encoding log data from events to groups and for various downstream tasks.

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

# A APPENDIX

## A.1 SUMMARY VECTOR VISUALIZATION

With VHM loss, we expect the summary vector representations for the real and fake data exhibit significantly different patterns. We visualize the learned summary vector representations using LLE, PCA and t-SNE in Figure 5 The results show evident separable clusters for real/fake data.

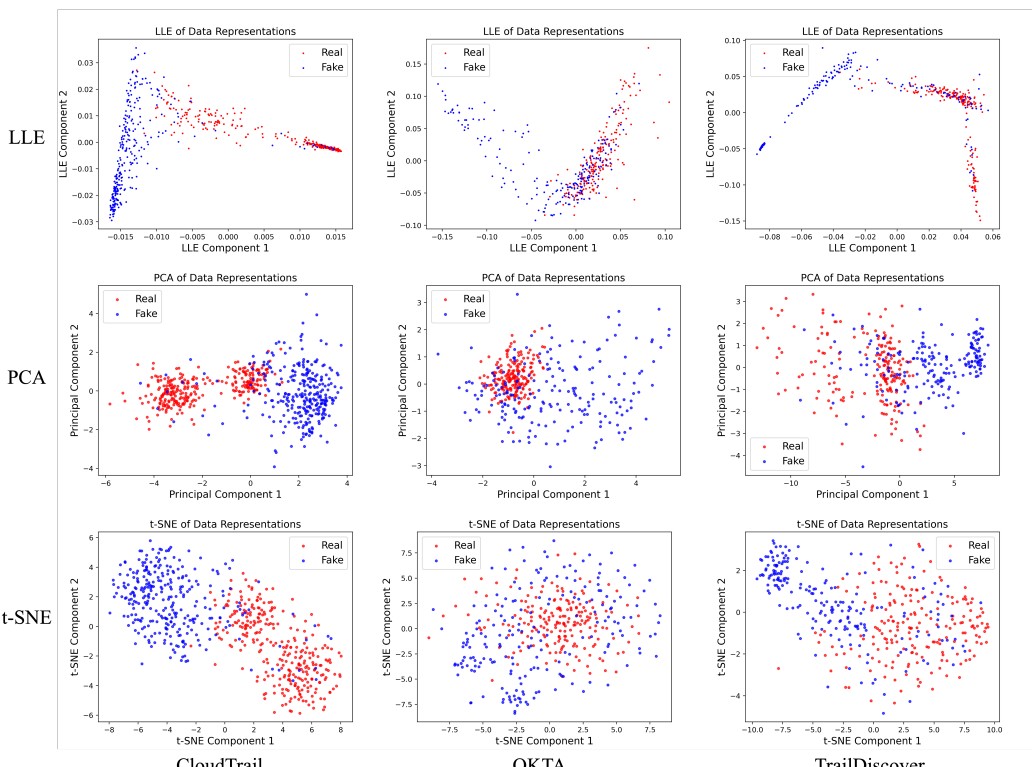

Figure 5: Visualization of summary vectors.

## A.2 TRAINING LOSS CURVE

To validate the efffectiveness of our hierarchical framework, we track the training/testing loss during the training in Figure 6. As can be observed in the curve, our hierarchical transformer exhibits faster convergence and better performance.

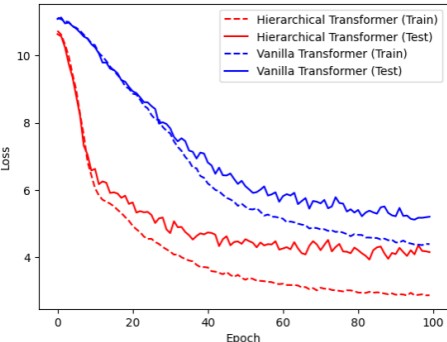

Figure 6: Loss curve during training.

## A.3 ANOMALY DETECTION BY FAKE RATE

Table 7 shows the significant differences in fake rates across different datasets under various $T$ values.

| Data | T=50 | T=20 | T=10 | T=5 | T=1 |
|------|------|------|------|-----|-----|
| Real | 4.08% | 11.27% | 18.03% | 31.23% | 73.54% |
| Fake | 32.72% | 45.36% | 55.94% | 68.73% | 86.65% |

Table 7: Fake rate of different datasets under various $T$.

By setting different thresholds $\alpha$, we can predict whether a log is fake or not, i.e., a fake rate $> \alpha$ indicates the log is fake. Consequently, we can calculate the accuracy for both real and fake logs separately and then compute their average to determine the overall accuracy of the model. With $T = 10$, we show the average accuracy at different threshold levels in Table 8. The results demonstrate that using the fake rate can achieve high accuracy (up to 95.96%) in synthetic anomaly detection.

| Threshold $\alpha$ | 25% | 30% | 35% | 40% |
|--------------------|-----|-----|-----|-----|
| **Accuracy** | 91.92% | 95.45% | 95.96% | 92.42% |

Table 8: Average accuracy at different threshold levels.

