# OpenReview forum: "HLogformer: A Hierarchical Transformer for Representing Log Data"
_ICLR.cc/2025/Workshop/BuildingTrust — Submitted to BuildingTrust_

### Official Review · Reviewer_kQy2 · 2025-02-16
**review from kQy2**

**Rating:** 6
**Confidence:** 4

**Review:**

The paper presents a sound approach to handling log data using a hierarchical transformer. The overall quality is solid, with extensive experimental evaluations across multiple datasets and tasks, which addresses both memory inefficiencies and the loss of contextual hierarchy. The pros and cons are listed below:

---
**Pros**:
* The focus on log data is interesting, and the plug-in nature of HLogformer for different backbones shows innovation.
* Tailored transformer design that leverages the inherent hierarchy of log data.
* Demonstrated reduction in memory consumption and improved performance across various tasks.

**Cons**:
* The paper sometimes reads as an incremental extension of existing hierarchical transformer ideas rather than a groundbreaking departure. It would be interesting to see more discussion on how this approach fundamentally differs from other hierarchical models. Furthermore, I'm curious about what the fundamental difference between log data and some other structured data like table, graph, etc., is. To me, some other methods that have shown good performances on table/graph may also work for this log data as sometimes this structured data is also represented as a hierarchical tree. The authors may provide more baselines or analysis for clarifying this issue to highlight the novelty of this paper.
* The bidirectional summary passing mechanism from lines 238 to 243 is not clear: what is the new proposed bidirectional summary passing technique? and how does this mechanism impact performance, especially in comparison to standard approaches?
* How would HLogformer perform on logs with less regular or evolving structures? I would like to know more about the authors' insights on how this method can be applied to real applications and what type of cases could benefit from this work?

Overall, while the paper makes a valuable contribution, clarifying some of its technical aspects and expanding the discussion on limitations would strengthen the work.

----
I have some further concerns regarding the topic of this paper; it may not fit the requirements of this workshop. As it mainly focuses on proposing a sound method for using a hierarchical transformer to handle log data, while the workshop's purpose is mainly to build trust in language models **(requires ACs to take a look at this issue)**.

---

### Official Review · Reviewer_eMUN · 2025-02-27
**Out of Scope of the Workshop**

**Rating:** 3
**Confidence:** 4

**Review:**

The authors propose a new transformer architecture, HLogFormer, capable of dealing with log data more efficiently and accurately than standard linear transformers.

The paper itself is interesting, and the HLogFormer can clearly have some targeted use in the real world. However, I fail to see how the paper aligns with the scope of the workshop. The paper focuses on developing a new transformer technique designed specifically to handle log data. This is, however, not a contribution on the trustworthiness of LLMs.

My rating is not a comment on the quality of the paper, but instead, on the lack of any connection between the scope of the workshop and the contributions of this paper.
Given the paper is completely out of scope of the workshop, I don't believe the reviewers should be expected to have any further comments on this paper.

---

### Official Review · Reviewer_HPDT · 2025-03-02
**The paper proposes a novel hierarchical transformer that better represents nested log data by exploiting its tree structure, thereby reducing memory usage and improving downstream tasks like anomaly detection and recommendations compared to transformers for log data.**

**Rating:** 6
**Confidence:** 3

**Review:**

This paper proposes HLogformer, a hierarchical transformer tailored for dictionary-like, nested log data. The authors identify a gap in how conventional transformers handle logs: treating them as linear sequences and consuming excessive memory. They propose a model that exploits the hierarchical structure of logs to reduce complexity and preserve context.

This work provides a well-reasoned solution for a notable domain problem. The discussion of how log entries can be viewed as a tree structure instead of a flat sequence makes sense, and it is applied in the paper with a step-by-step hierarchical summary framework. The clarity of the text is also good. The authors carefully motivate the challenge of dealing with dictionary-like data, show how they derive a hierarchical tree representation, and then detail their transformer architecture that processes segments in a way that reflects this inherent nesting. It is a logical progression, and I seldom felt lost while reading. The originality of the approach lies in how the authors use progressive summaries that pass from lower-level segments of the log to higher-level segments, ensuring that both fine-grained and broad contextual cues can be captured.

Pros include the strong argument about the dictionary-like structure in logs, the consistent demonstration of memory savings, and the application of the approach to multiple tasks (anomaly detection, recommendation, etc.) to prove wide coverage. Another plus is that it fits with a variety of frameworks, so it is generally reusable as a plugin for hierarchical log representation. As for con, I noticed that the text occasionally lacks further elaboration on hyperparameter tuning or training details that might influence reproducibility.

One thing I'm less certain about the fit of HLogformer with the topic of the workshop, which centers around methods to build trust in Large Language Models, while this work, albeit interesting, focuses more on proposing a variants of transformer for a different data modality.

---

### Decision · Program_Chairs · 2025-03-04

Reject